# Parenteral Nutrition in Patients with Incurable Cancer: Exploring the Heterogenous and Non-Randomised Clinical Landscape

**DOI:** 10.3390/curroncol32110644

**Published:** 2025-11-18

**Authors:** Marianne Erichsen, Tora S. Solheim, Inger Ottestad, Ingvild Paur, Rikka F. Sande, Astrid Nygaard, Emilie H. Markhus, Lene Thoresen, Morten Thronæs, Randi J. Tangvik, Kari Sygnestveit, Patrik Hansson, Cathrine Vestnor, Gunnhild Jakobsen, Ørnulf Paulsen, Erik Torbjørn Løhre, Trude R. Balstad

**Affiliations:** 1Department of Clinical Medicine, Clinical Nutrition Research Group, UiT, The Arctic University of Norway, 9019 Tromsø, Norway; ingvild.paur@uit.no (I.P.); kari.sygnestveit@helse-bergen.no (K.S.); patrik.hansson@uit.no (P.H.); 2Cancer Clinic, St. Olavs Hospital, Trondheim University Hospital, 7030 Trondheim, Norway; tora.s.solheim@ntnu.no (T.S.S.); l-thores@online.no (L.T.); morten.thrones@stolav.no (M.T.); gunnhild.jakobsen@stolav.no (G.J.); erik.t.lohre@ntnu.no (E.T.L.); 3Department of Clinical and Molecular Medicine, Faculty of Medicine and Health Sciences, NTNU—Norwegian University of Science and Technology, 7030 Trondheim, Norway; 4Department of Nutrition, Institute of Basic Medical Sciences, Faculty of Medicine, University of Oslo, 0313 Oslo, Norway; i.o.ottestad@medisin.uio.no; 5The Clinical Nutrition Outpatient Clinic, Section of Clinical Nutrition, Department of Clinical Service, Division of Cancer Medicine, Oslo University Hospital, 0424 Oslo, Norway; 6Norwegian National Network for Disease-Related Undernutrition, and Section of Clinical Nutrition, Department of Clinical Service, Oslo University Hospital, 0424 Oslo, Norway; 7Norsk Helsenett SF, 7030 Trondheim, Norway; rikka.sande@nhn.no; 8Cancer and Hematology Center, Vestfold Hospital Trust, 3103 Tønsberg, Norway; astrid.signe.nygard@larvik.kommune.no; 9Clinical Nutrition Center, Haukeland University Hospital, 5007 Bergen, Norway; emilie.hjonnevag.markhus@helse-bergen.no; 10Centre for Nutrition, Department of Clinical Medicine, Faculty of Medicine, University of Bergen, 5007 Bergen, Norway; randi.tangvik@uib.no; 11Centre for Crisis Psychology, Faculty of Psychology, University of Bergen, 5007 Bergen, Norway; 12Department of Research and Development, Haukeland University Hospital, 5009 Bergen, Norway; 13Department of Food and Nutrition and Sport Science, University of Gothenburg, 40530 Gothenburg, Sweden; 14Department of Public Health and Nursing, NTNU—Norwegian University of Science and Technology, 7030 Trondheim, Norway; cathrinev93@hotmail.com; 15Palliative Care Unit, Department of Medicine, Telemark Hospital Trust, 3710 Skien, Norway; paor@sthf.no; 16European Palliative Care Research Centre (PRC), Department of Oncology, Oslo University Hospital, 0424 Oslo, Norway; 17Institute of Clinical Medicine, University of Oslo, 0372 Oslo, Norway

**Keywords:** parenteral nutrition, palliative cancer care, real-world data

## Abstract

Parenteral nutrition (PN) is a controversial and understudied topic in palliative care. Few studies explore how PN is administered to patients with incurable cancer, and strict patient selection in clinical trials often excludes those commonly treated in real-world practice. This multicentre study provides valuable insights into the everyday clinical use of PN. Common reasons for initiating PN were eating difficulties or obstructions in the digestive tract, with PN doses often administered below estimated needs and requiring frequent adjustments. Few patients received anticancer treatment, though some used PN as a bridge to future therapies; however, overall survival was short. This study highlights the importance of individualised PN treatment, carefully and safely managed to meet the patients’ palliative care situation.

## 1. Introduction

Patients with incurable cancer may experience a multitude of symptoms and gastrointestinal (GI) obstruction, resulting in critically low food intake and life-threatening malnutrition [1,2,3]. For these patients, parenteral nutrition (PN) may be indicated [4]. In cases where patients with progressive cancer undergo a gradual decline in weight and appetite, with no detectable GI organ malfunction, the role of PN is more unclear [4,5].

A limited number of studies have addressed the efficacy of PN on clinical outcomes in this patient group [6,7]. Hence, current international guidelines are based on sparse evidence and lack detailed recommendations on indications, administration, dosage, and duration of the use of PN for these patients [4,5,8,9]. Determining the optimal use of PN in patients with incurable cancer is challenging [4,5,8,9].

Previously conducted randomised controlled trials (RCTs) on PN in incurable cancer provided evidence for the effect of PN on chosen outcomes [7,10,11]. However, these studies included small groups of highly selected patients and strict eligibility criteria, leading to selection bias and limiting the external validity. This may contribute to a knowledge gap between a controlled trial environment and the complexities of real-world clinical practice. Additionally, the RCT study design may also lead to recruitment challenges, resulting in a limited number of partly underpowered studies on PN for patients with incurable cancers [7,10,11]. With a paucity of strong scientific evidence, clinical practice will be governed by clinical evidence, user perspectives, and local traditions [12].

Real-world data are recognised as a valuable complement to RCTs in healthcare decision-making [13]. Data derived from clinical practice reflect cancer patient populations with varying prognoses and treatment patterns, offering insights into long-term safety, adherence, and overall impact. To guide the development of future high-quality trials, real-world data can offer new insights into the practical administration of PN treatment to patients in everyday clinical practice [14]. The overall aim of this study was to describe the practices and actual outcomes of parenteral nutrition in patients with incurable cancer at Norwegian hospitals. Therefore, we conducted a multicentre retrospective study to investigate key aspects of PN treatment, such as indications, doses and duration, complications, benefits, and survival.

## 2. Materials and Methods

### 2.1. Study Design and Population

This multicentre cohort study consisted of data retrospectively collected from medical records (Figure 1). Data were collected from patients treated at two university hospitals and two local hospitals in Norway between January 2011 and December 2017. The included hospitals were St. Olavs University Hospital, Trondheim; Haukeland University Hospital, Bergen; Telemark Hospital Trust, Skien; and Vestfold Hospital Trust, Tønsberg.

Patients were identified exclusively through electronic hospital charts, which included referral records, medical equipment databases, and procedure codes for PN. We included adult deceased patients (>18 years) with incurable locally advanced or metastatic cancer. For patients who had received PN on multiple occasions, information from the last PN treatment period prior to death was collected. A break ≤ 14 days was considered a treatment pause, while resumed PN treatment after a break > 14 days was considered a new treatment period.

### 2.2. Data Collection

Data were collected from PN initiation until death. Baseline characteristics included demographics (age, sex), description of the cancer and treatment (cancer diagnosis, metastasis, current and previous anticancer treatment), comorbidities and other medications, physical function as assessed by the Eastern Cooperative Oncology Group (ECOG), Body Mass Index (BMI) (kg/m^2^), weight loss (%) in the last two weeks to six months, ascites, and biochemical blood analysis—including C-reactive protein (CRP) (mg/L) and albumin (g/L). The main care provider was defined as the institution where patients received most of their PN treatment. Descriptions of nutrition-related data and PN treatment at baseline included previous use of PN, dietary intake records, estimated energy requirements, indications, and dose ordinations. From the start of PN until death, data were collected on oral intake or tube feeding complementary to PN, descriptions of PN treatment such as method of administration (e.g., catheter type and solutions/additives), dose (ordinated and given), and dose adjustments, duration, and pauses (number and duration), observed benefits (e.g., increased feelings of hope or improved wellbeing) and complications (e.g., nausea, oedema, infections) as described in the journals of patients, reasons for discontinuation, anticancer treatments (previous, newly initiated or resumed, and discontinued), as well as survival from the start of PN and after discontinuation. Additional information and comments from data abstractors were reported as free text and included benefits, complications, reasons for discontinuation of PN (not covered by pre-defined response options), and free-text comments.

In medical chart reviews, it is recommended to define study variables and inclusion/exclusion criteria in advance, as well as establish clear procedures for handling missing data [15,16]. A detailed research manual was developed describing all processes from patient identification and eligibility assessment to variable definitions and data collection procedures, ensuring standardised and reliable data collection. Data abstractors were trained, and data were monitored during and after data collection. Identified outliers or errors were addressed and corrected prior to the conclusion of the study.

### 2.3. Ethics

The study was performed in accordance with the Declaration of Helsinki and was approved by the Norwegian Regional Committee for Health and Research Ethics (REK) (25062/REK Mid. 10 October 2018). REK provided an exemption to obtaining informed consent from relatives of the deceased patients. The study was registered at Clinicaltrials.gov (ID NCT04456647).

### 2.4. Statistical Analyses

Continuous variables are presented as means with standard deviations (SD) if normally distributed and, otherwise, presented as medians with interquartile ranges (IQR) or range. Dichotomous variables are expressed as frequencies and percentages. The median and maximum doses of PN administered during treatment were calculated based on doses given and dose adjustments for each individual. Free-text responses were manually analysed using thematic analysis to identify recurring themes [17]. A number of observations are reported for each variable, and no data imputation was performed. Data were analysed using R software (R version 4.3.1) and figures were created using Adobe Illustrator^®^(version C6S).

## 3. Results

A total of 507 patients were included in the analysis (Table 1), of whom 197 (39%) were recruited from St. Olavs University Hospital, Trondheim, 133 (26%) from Haukeland University Hospital, Bergen, 98 (19%) from Telemark Hospital Trust, Skien, and 79 (16%) from Vestfold Hospital Trust, Tønsberg.

### 3.1. Patient Characteristics

Table 1 and Appendix A describe patient characteristics. The mean age (SD) was 65 (13) years, 53% were women, and the mean (SD) Body Mass Index (BMI) was 21.4 (4.2) kg/m^2^. Three months prior to PN treatment, 48% of the patients experienced a mean weight loss of more than 10%, and 42 (8%) maintained or gained weight. At the start of PN treatment, 40% of the patients had an Eastern Cooperative Oncology Group (ECOG) performance status between one and three, 30% had ascites, and 36% did not have any comorbidities.

Median (IQR) time since diagnosis was 11 (3–24) months. The most common cancer diagnoses were upper GI tract (37%), colorectal (21%), and gynaecological cancer (14%). A total of 84% of the patients had metastatic disease. Two-thirds of the patients received no anticancer treatment at the initiation of PN, and 12% had received no anticancer treatment at all.

### 3.2. Nutrition-Related Data

Calculated energy intake from dietary registrations (*n* = 95) prior to PN start was mean (SD) 804 (163) kilocalories per day (kcal/day) (median (range) registration of 2 (1–9) days). Mean (SD) estimated energy requirement before PN (*n* = 250) was 1973 (357) kcal/day. A clinical dietitian was involved in the care of 46% of the patients.

All included patients started PN treatment during their hospital admission. During the period patients received PN, 66% had hospital as the main care provider, both in- and outpatient care, while the remaining patients received treatment with home care services or nursing homes/care facilities as the main care provider.

### 3.3. Indication, Administration, and Discontinuation of Parenteral Nutrition

#### 3.3.1. Indications for Parenteral Nutrition

Two indications for PN treatment dominated, “insufficient oral intake and/or tube feeding” (75%) and/or “gastrointestinal malfunction” (47%), and both were frequently observed in combination with other indications (Table 2). One single indication for starting PN was registered for 52% of the patients, 47% had two or three indications, and four patients had four indications. PN was administered to facilitate initiation or bridging of anticancer treatment for 43 patients (8%), of whom 15 (35%) received subsequent anticancer treatment. The indication “patient wish” (4%) was often combined with other indications, most often with “insufficient oral intake and/or tube feeding”.

#### 3.3.2. Dosages and Infusion Rate

Table 2 presents details of the PN doses administered to patients. The median (IQR) starting PN dose was 1000 (550–1100) kcal/day, and the median (IQR) dose administered during PN treatment was 1050 (825–1125) kcal/day. Only 7% of the patients received a median dose of 1600 kcal or more. The median dose administered during treatment at a group level corresponded to 53% of estimated energy requirements. The most common doses delivered were 550 kcal, 1000 kcal, 1100 kcal, and 1600 kcal, corresponding to the sizes of multichambered PN bags, not necessarily considering the patient’s body weight (Figure 2A). Doses were adjusted median (IQR) 3 (1–5) times during treatment, and most often the doses were increased (Figure 2B). The median infusion rate was 75 mL/h, equivalent to delivering ~1100 kcal over 14 hours. Details of the administration of PN, such as route of infusion, infusion schedule, PN solution, and addition of vitamins, minerals and additional fluids, are presented in the Appendix A.

#### 3.3.3. Pauses and Duration

Treatment pauses occurred in 41% of the patients, with the number of pauses ranging from one to six and most commonly with a median (IQR) duration of 2 (1–5) days (Figure 2C). The duration of PN treatment ranged from 1 to 1060 days, with a median (IQR) duration of 34 (13–84) days. Patients who lived shorter than three months had a median duration of PN of 26 days, while those surviving over three months had a median duration of PN of 96 days. Most patients received PN treatment daily (73%), while 11% received it one to three days per week, and 11% four to six days per week (Figure 2D).

#### 3.3.4. Complementary Intake

Most patients (94%) had complementary intake of solid foods, caloric liquids, oral nutrition supplements, or tube feeding, or they received supplementation of intravenous glucose (Table 2). Dietary intake records during PN were performed for 37% of the patients, with a median registration period of 4 days. Energy calculations were not recorded.

#### 3.3.5. Discontinuation of Parenteral Nutrition

The most frequent reason for discontinuation of PN was the expected imminent death of the patient (47%), underpinned by a median survival of seven days after discontinuation of PN. Other common reasons included converting to oral intake or tube feeding (23%) and complications associated with PN (17%). Additionally, 5% of the patients wished to discontinue treatment due to the burden (Table 2). Patients can experience the constraints imposed by technicalities such as catheters and slow infusion rates as burdens of treatment.

### 3.4. Disease and Treatment-Related Complications and Benefits of Parenteral Nutrition Treatment

Nausea (52%), vomiting (46%), and oedema (37%) were the most frequently reported complications during PN treatment (Table 3). However, for 15% of the patients, improved wellbeing was reported, suggesting a potential benefit of PN. Other positive observations included reduced stress related to food intake (3%) and increased feelings of hope (1%).

### 3.5. Survival

Survival from the start of PN treatment was median (IQR) 70 (33–153) days, whereof 42% of the patients lived after three months. After discontinuation of PN, survival was median (IQR) 7 (1–40) days.

## 4. Discussion

### 4.1. Main Findings

We provide a comprehensive real-world perspective on how PN treatment is administrated to patients with incurable cancer. We report four important findings: First, PN is primarily initiated in response to insufficient intake, regardless of whether the patients have gastrointestinal malfunction, ascites, or established weight loss. Second, one-third of the patients who received bridging PN were later able to receive anticancer treatment; however, their median survival was short. Third, PN treatment is most often administered as a supplement, frequently tailored to meet the individual’s palliative care situation. Finally, for most patients, PN was continued until death, potentially highlighting the difficulties of withdrawing treatment when death is inevitable.

### 4.2. Clinical Implications

Interventions intended to cover nutritional requirements are recommended in patients with incurable cancer who have an expected survival exceeding two to three months [4,5]. In our study, PN was mostly initiated due to critically low food intake, alone, or in combination with other indications or criteria. In a previous RCT investigating the effects of PN, weight loss was an inclusion criterion [7]. Although most patients in our cohort experienced weight loss prior to PN start, notably, half of the patients reported weight loss of 10% or more. However, for a small group, no weight loss or even weight gain was observed. Limiting enrolment to patients with weight loss may exclude patients experiencing ascites or oedema, who represent at least 30% of the patients in this real-world population. Despite the high prevalence, ascites is rarely reported in PN studies [6,18], and it is even used as an exclusion criterion [7,19,20]. While high-protein nutrition is recommended for patients with ascites, there is sparse evidence regarding the optimal dose for malnourished patients with advanced cancer [21]. A previous study comparing high-protein PN and standard PN found that the high-protein solution was a safe and at least as an effective option for palliative cancer patients [22]. Interestingly, none of the patients in our study received PN with higher protein content than the standard solutions. Future studies should explore the effect of high-protein solutions versus standard solutions on muscle mass, ascites, and total symptom burden. Additionally, the use of weight loss as a central inclusion criterion in clinical trials should be reconsidered, particularly without accounting for fluid accumulation. Conditions like oedema and ascites, which are common in this population, can worsen the nutritional symptom burden and necessitate the initiation of PN when other interventions are ineffective or inapplicable.

Clinical guidelines recommend nutritional support as an integral component of anticancer treatment, recognizing that weight loss significantly diminishes tolerance to anticancer therapies [4]. Despite the fact that fewer than 10% of the patients received PN to enhance their ability to tolerate subsequent treatments, this finding is particularly noteworthy, highlighting an area that has been underexplored in RCTs [7,10,11]. In our study, approximately one-third of these patients successfully resumed anticancer treatment, and they exhibited better performance status at the initiation of PN and had a median survival exceeding three months. Those who resumed anticancer treatment were most often followed by a clinical dietitian and received slightly higher median and maximum PN doses compared with those who did not resume treatment, with half experiencing no potential PN-related complications. These findings suggest that PN may offer benefits to a selected group of reasonably fit patients, potentially enhancing their ability to tolerate subsequent life-prolonging cancer therapies.

Our findings highlight the variability in PN treatment among patients with incurable cancer. Half of the patients received PN daily in doses lower than their estimated energy requirements. Higher doses of PN (≥1600 kcal/day) were commonly administered for shorter periods, and a few of the patients (7%) received doses corresponding to estimated energy requirements for the majority of their PN treatment period. Patients who were followed by a clinical dietitian most often had their energy requirements estimated, received PN for longer periods of time, and had the PN dose adjusted more frequently. These patients received higher maximum doses of PN; however, the median dose of PN received during treatment did not differ. Previous studies often report the planned PN dose based on estimated energy requirements, but very few specify the dose administered or details on how the PN treatment is adjusted or paused over time [23,24,25,26,27]. This leads to a gap in the understanding of the individual tolerability of PN treatment since details of the actual administration are lacking. We found that PN mainly provided support to patients by reducing the gap between patients’ energy intake from food, drinks, and ONS and their nutritional requirements, with treatment tailored to an individual’s palliative situation. We also found that patients with gastrointestinal malfuction typically had fewer treatment pauses and usually received PN daily since their capability of complementary intake was low. Given the challenges of estimating energy needs and uncertainties regarding nutrient utilization at this stage of disease, administering PN as a supplement based on standardised bag sizes, rather than strict calculations, may be a reasonable patient-centred approach. This pragmatic strategy allows for flexibility in treatment, accommodating patients’ preferences and clinical situations, while enabling pauses for meaningful daily activities.

Duration of PN treatment varied widely among patients. Notably, 25% of the patients were able to resume oral food intake or tube feeding, most often due to recovery of gastrointestinal function or regaining appetite after a period of PN. However, most resumed treatment until imminent death. Current guidelines recommend to discontinue treatment when the burden outweighs the benefit [4]. In our study, some patients stopped PN due to complications, lack of perceived benefit, or personal choice. These findings highlight that while PN can temporarily bridge enteral intake for some, determining the appropriate time for discontinuation is a composed evaluation based on the clinical symptoms, prognosis, and preferences of each patient.

The documented benefits of PN included improved wellbeing, reduced eating-related stress, and increased hope, though these were recorded in only one-fourth of the patients. A recent secondary analysis from a prospective cohort study of palliative cancer patients found an association between PN and improved quality of life (QoL) when PN was initiated early and in accordance with guidelines [28]. Our findings, along with previous research, underscore the need for further exploration of the role of PN in end-of-life care on outcomes that matter for patients.

The clinical insight from our study is that the palliative care pathways vary markedly between patients. Our findings suggest that healthcare providers strive to tailor treatment to the individual palliative care needs of each patient, often with the involvement of clinical dietitians to optimise nutritional management and balance the potential benefits and burdens of PN.

Given the role of patients and their relatives in decisions regarding the initiation and discontinuation of treatment, the anticipated benefits, such as anticancer treatment tolerance, and the possible complications of PN should be thoroughly evaluated and transparently discussed throughout treatment. These discussions should align with the principles of shared decision-making and advanced care planning, ensuring that care is patient-centred and tailoring individual preferences and prognosis [29].

### 4.3. Comparison with Previous Studies

A few RCTs have investigated the effect of PN on selected outcomes such as muscle mass and various domains of QoL. However, these studies often impose strict eligibility criteria to minimise bias and confounding, leading to the exclusion of a significant portion of the population encountered in clinical practice. To exemplify this, the trial by Oh et al. [11] excluded patients capable of tube feeding and those with a life expectancy beyond 12 weeks. Bouleous et al. [7] required specific thresholds for BMI or weight loss according to malnutrition criteria, functional gastrointestinal tract without symptomatic peritoneal carcinomatosis, and set survival limits of under 12 weeks or of longer than two months. Similarly, Obling et al. [10] restricted inclusion for patients with GI-cancers who had an ECOG performance status of 0 to 2. Applying similarly stringent selection criteria, 80% of our patients would not have been eligible for inclusion. Furthermore, exclusion based on conditions such as functional or actual short bowel syndrome, poorly controlled diabetes, severe ascites, or symptomatic peritoneal carcinomatosis would have further reduced the number of eligible patients [7,10,11,20]. This gap between clinical trial populations and real-world settings limits the generalisability of their findings to everyday clinical practice.

### 4.4. Future Research

To advance knowledge and bridge the gap between current trials and clinical practice, pragmatic randomised controlled trials or prospective interventions should be conducted to evaluate the effectiveness and potential adverse effects of different PN formulations (e.g., standard vs. high-protein solutions) and doses on clinical outcomes [30,31]. These trials should be designed in accordance with guidelines, where nutritional interventions are initiated in a stepwise manner for patients with functional GI-tracts [4,5,9]. Trials need to reflect the heterogenic patient population in real-world clinical settings and prioritise a patient-centred approach by considering individual needs, preferences, and quality of life domains as key outcomes [30,31].

### 4.5. Strengths and Limitations

To our knowledge, no other studies have provided this level of detail on how PN is administrated and adjusted to patients with incurable cancer. The inclusion of data only from Norwegian Hospitals in addition to the lack of studies with similarly detailed PN data from other countries both limit the opportunity for cross-national comparisons and challenge external generalisability evaluation. However, including all patients receiving PN across multiple hospitals enables the inclusion of a diverse patient population that reflects real-world clinical scenarios, thereby enhancing the external validity and complementing the findings from RCTs. The study spanned seven years and encompassed a large sample size. Conducting a prospective study of this scale would be highly time-consuming and could introduce selection bias, potentially favouring the inclusion of the healthiest and most motivated patients.

A key limitation of retrospective studies is the reliance of clinical data not originally intended for research, inevitably leading to missing information [32]. Incomplete documentation may lead to a skewed understanding of the appropriateness of treatment. This includes gaps in registrations on perceived benefits or side effects of PN. Additionally, factors such as minor changes in a patient’s health condition or psychosocial situation influencing PN treatment may not be recorded [33]. While standardised data like cancer diagnoses and blood analyses were well documented, PN administration records were variable due to non-standardised documentation practices. Having limited data on oral energy intake, tube feeding, and glucose IV alongside PN restricts interpretation of total energy intake. Only the calculated oral intake before the start of PN, if oral intake or tube feeding was given complementary to PN, and the starting dose of glucose IV were recorded. Nevertheless, dietary registrations can also vary, are prone to underreporting, and may not accurately reflect actual energy intake over time.

Finally, using data from 2011–2017 means our findings are contextualised within a historical therapeutic landscape which has since evolved considerably [34]. While our cohort had some exposure to novel agents (four patients received immunotherapy; eighteen received targeted therapy), these treatments are now far more standard. New systemic treatments might improve cancer outcomes today; however, these same treatments, e.g., immunotherapies, can induce severe gastrointestinal side effects such as colitis, which might also lead to an increased demand for supportive treatments like PN [35]. Despite the limitations discussed, this study provides a comprehensive view of PN treatment within this specific setting. The evolution of modern oncology requires updated, prospective data to accurately assess how new treatment paradigms currently impact the need for PN and the associated outcomes.

## 5. Conclusions

Our study provides a real-world perspective on the use of PN in patients with incurable cancer, of whom up to 80% would not meet the eligibility criteria of previous trials. Current international guidelines are therefore based on sparse evidence and lack detailed recommendations for PN in real-world clinical populations. We found that PN is often initiated in response to insufficient intake, regardless of gastrointestinal malfunction, ascites, or weight loss and is frequently tailored to the palliative care context. Additionally, PN may benefit selected patients by improving their ability to tolerate anticancer treatments. Regular evaluation and documentation of the perceived benefits and burdens of PN are essential to guide decisions on timely discontinuation. To bridge the gap between the current literature and clinical practice, pragmatic real-world trials should be conducted to evaluate the effectiveness of PN in the care of the patients who actually receive it in clinical practice. Studies avoiding strict eligibility criteria and providing nutritional interventions in a stepwise manner in accordance with current guidelines should be conducted to evaluate the benefits and symptom burden of PN while focusing on patient-centred outcomes.

## Figures and Tables

**Figure 1 curroncol-32-00644-f001:**
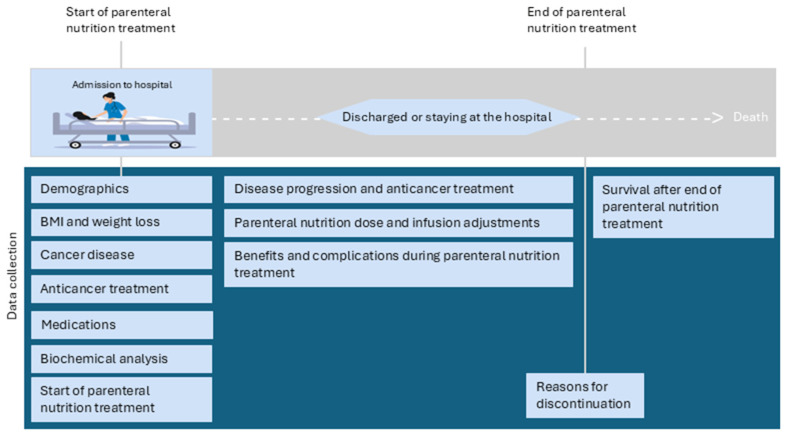
Overview of the data collection in the PATNIC study.

**Figure 2 curroncol-32-00644-f002:**
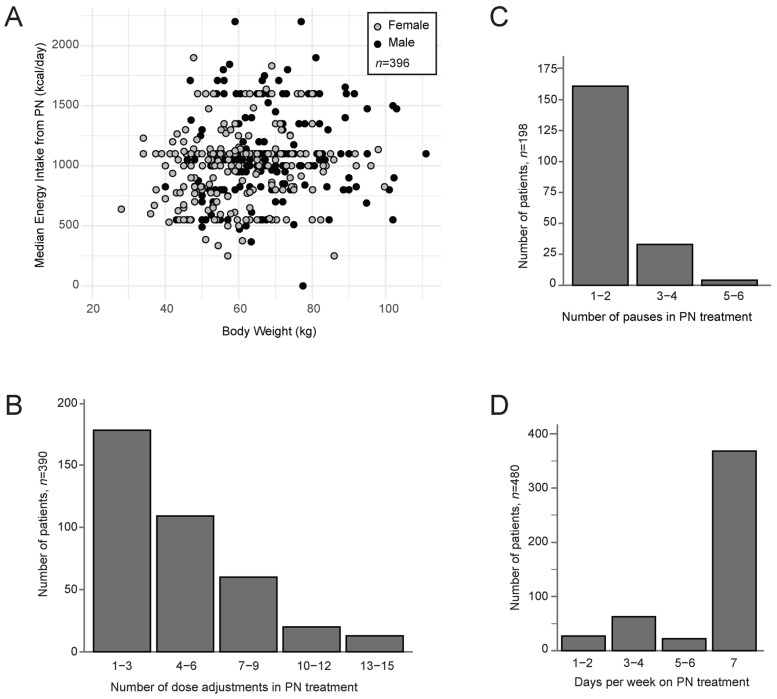
(**A**) Relationship between body weight and median energy (calorie) intake provided through parenteral nutrition (PN). (**B**) Number of adjustments in PN treatment dose. (**C**) Number of pauses in PN treatment. (**D**) Number of days a week receiving PN treatment.

**Table 1 curroncol-32-00644-t001:** Patient characteristics at the start of parenteral nutrition.

Descriptive Characteristics	*n* = 507, *n* (%)	Mean (SD) or Median (IQR)
Age, years, mean (SD)		65 (13)
Sex, females	270 (53%)	
Time since diagnosis, months, mean (SD)	507 (100%)	11 (3–24)
BMI, kg/m^2^, mean (SD) ^1^	389 (77%)	21.4 (4.2)
Height, cm, mean (SD)	464 (92%)	171.6 (8.9)
Weight, kg, mean (SD)	400 (79%)	63.1 (13.9)
**Weight loss, % ^2^**		
2 weeks–1 month, mean (SD)	146 (29%)	6.5 (4.0)
2–3 months, mean (SD)	169 (33%)	10.6 (6.0)
4–6 months, mean (SD)	173 (34%)	12.6 (6.7)
Albumin, g/L, mean (SD)	438 (86%)	31.9 (6.8)
CRP, mg/L, median (IQR)	487 (96%)	52 (17–107)
**ECOG (0–4) ^3^**	230 (45%)	
0	5 (<1%)	
1	40 (8%)	
2	88 (17%)	
3	77 (15%)	
4	20 (4%)	
**Cancer diagnosis**	507 (100%)	
Upper gastrointestinal tract ^4^	188 (37%)	
Colorectal	109 (21%)	
Gynaecological	69 (14%)	
Lymphoma	22 (4%)	
Lung	21 (4%)	
Head and neck	16 (3%)	
Breast	15 (3%)	
Prostate	9 (2%)	
Other ^5^	62 (12%)	
Metastasis		
Metastatic cancer (yes)	426 (84%)	
**Current anticancer treatment ^6,7^**	506 (100%)	
Systemic therapy ^8^	148 (29%)	
Radiotherapy	18 (4%)	
None	340 (67%)	
**Survival**		
Survival from start of PN, days, median (IQR)	507 (100%)	70 (33–153)
Survival after discontinuation of PN, days, median (IQR)	507 (100%)	7 (1–40)
Main provision of care ^9^		
Hospital/palliative care unit	332 (66%)	
Home	151 (30%)	
Other	24 (4%)	

PN = parenteral nutrition. SD = standard deviation. IQR = interquartile range. *n* indicates number of cases. ^1^ BMI = Body Mass Index. ^2^ Estimated as weeks to months prior to initiation of PN; missing for *n* = 201. *n* = 42 patients maintained or gained weight. ^3^ Missing for *n* = 264. ^4^ Oesophageal cancer, stomach cancer, pancreatic cancer, liver cancer, gallbladder cancer. ^5^ Skin 8 (<2%), Kidney 8 (<2%), Bladder 8 (<2%), Bone 6 (1%), Adrenal gland 5 (1%), Leukaemia 4 (<1%), Neuroendocrine 3 (<1%), Other 20 (4%). ^6^ Combination of several treatments possible. ^7^ Information not found for *n* = 1. ^8^ Chemotherapy 126 (25%), Immunotherapy 4 (<1%), Targeted therapy 18 (4%). ^9^ Main care provider is the institution that administered the patient follow-up of PN treatment.

**Table 2 curroncol-32-00644-t002:** Indication, administration, and discontinuation of parenteral nutrition.

Variable	*n* = 507, *n* (%)	Median (Range)
**Indication for start of parenteral nutrition ^1^**		
Insufficient oral intake or tube feeding	378 (75%)	
Gastrointestinal malfunction ^2^	237 (47%)	
Promote tolerance for anticancer treatment	43 (8%)	
Patient wish	20 (4%)	
Other	119 (23%)	
**Parenteral nutrition treatment**		
Starting dose (kcal/day)	493 (97%)	1000 (200–2200)
Starting infusion rate, mL/h	131 (26%)	75 (25–150)
Median dose (kcal/day)	497 (98%)	1050 (0–2200)
Max dose (kcal/day)	497 (98%)	1600 (1100–2900)
Pauses in treatment (yes) ^3^	208 (41%)	
Duration of pauses, days ^3^		2 (1–5) ^4^
Duration of treatment (days)	507 (100%)	34 (13–84) ^4^
**Complementary energy intake**		
Complementary oral/tube feeding/IV energy intake	475 (94%)	
Food intake	392 (78%)	
Caloric liquids	243 (48%)	
Oral nutritional supplements	240 (47%)	
Tube feeding	32 (6%)	
IV glucose	113 (22%)	
No oral/tube feeding or IV intake	32 (6%)	
**Reasons for discontinuation ^5^**		
Patient is terminal	238 (47%)	
Complications related to PN treatment	87 (17%)	
Transition to oral nutritional intake	86 (17%)	
No perceived benefits	57 (11%)	
Transition to tube feeding	29 (6%)	
Recovery of gastrointestinal tract functions	29 (6%)	
Patient wish due to burden of PN treatment	23 (5%)	
Other ^6^	19 (4%)	
Unknown	32 (6%)	

PN = parenteral nutrition; *n* indicates number of cases; kcal = kilocalories. ^1,5^ Several indications or reasons for discontinuation possible. ^2^ Perforation, intestinal obstruction or chylothorax, high-throughput entero-cutaneous fistulas, paralytic ileus, digestive haemorrhage, insufficient absorptive surface due to cancer surgery or radiation enteritis. ^3^ Defined as a break < 14 days. ^4^ Median (interquartile range). ^6^ Hospital discharge or patient wish.

**Table 3 curroncol-32-00644-t003:** Disease and treatment-related observations during parenteral nutrition treatment.

	*n* = 507, *n* (%)
**Reported complications during PN ^1^**	
Nausea	262 (52%)
Vomiting	235 (46%)
Oedema	187 (37%)
Dyspnoea	154 (30%)
Ascites	142 (28%)
Infections ^2^	122 (24%)
Diarrhoea	109 (22%)
Elevated liver enzymes	108 (21%)
Tachycardia	61 (12%)
Feeling cold	58 (12%)
Hypotension	42 (8%)
Dizziness	31 (6%)
Sepsis ^2^	29 (6%)
Feeling warm	14 (3%)
Hypertension	17 (3%)
Headache	7 (1%)
Thrombophlebitis	5 (1%)
No complications reported	74 (15%)
Other ^3^	233 (46%)
**Consequences or reactions to complications ^4^**	
Pause in PN treatment	54 (10%)
Reduction in PN infusion or dose	34 (7%)
Termination of PN	22 (4%)
Diuretic drugs	56 (11%)
Drainage of accumulated liquid	70 (14%)
Other medical interventions ^5^	29 (6%)
None	235 (46%)
**Positive observations reported in relation to PN ^6^**	
Increased wellbeing	76 (15%)
Reduced stress related to food intake	16 (3%)
Increased hope	3 (<1%)
Other ^7^	41 (8%)
Not reported ^8^	380 (75%)

*n* indicates number of cases. ^1^ Several complications could be registered for the same patient. ^2^ Any infections. ^3^ E.g., abdominal pain, anxiety, electrolyte disturbances, fever, kidney failure, dry mouth, obstipation, pleural effusion. ^4^ Several consequences could be registered for the same patient. ^5^ E.g., antiemetics, antibiotics. ^6^ Several benefits could be observed for the same patient. ^7^ E.g., increased appetite, stable weight. ^8^ No observations reported in relation to PN treatment.

## Data Availability

The data presented in this study are available on request from the corresponding author.

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
