# Peer review of "Parenteral Nutrition in Patients with Incurable Cancer: Exploring the Heterogenous and Non-Randomised Clinical Landscape"

_curroncol, 2025, doi:10.3390/curroncol32110644_

Round 1

Reviewer 1 Report

Comments and Suggestions for Authors

This study investigates the real-world practice of parenteral nutrition (PN) for cancer patients and may be significant in terms of providing insight into the current situation.  However, I would like to request that the following points be reconsidered and revised.

Major critique:

  • First, I would like to see a clear message from this paper about what readers can apply to future clinical practice. The indications for PN in clinical practice do not always follow guidelines, but are influenced by the factors such as the continuation of chemotherapy and the wishes of the patient and their family.  I would like to see the authors provide more support and decision-making regarding whether the findings of this study are still recommended and how they differ from the guidelines.
  • The study has included cases from 2011 to 2017, so the data do not reflect subsequent recent advances in cancer chemotherapy and treatment systems. I would also like to see the authors consider the possibility that the situation may be even different today, given the emergence of ICIs.
  • Clinical dietitians were reported to be involved in the care in 46% of cases (P6, L1-2). I would like to see a comparison of PN implementation situation between those with and without such involvement by dieticians (sub-analysis).  This could provide valuable data that will shed light on the role of registered dietitians in nutritional management.

Author Response

Comment 1: [This study investigates the real-world practice of parenteral nutrition (PN) for cancer patients and may be significant in terms of providing insight into the current situation.  However, I would like to request that the following points be reconsidered and revised.]

Response 1: [Thank you very much for taking the time to review this manuscript. We believe that your suggestions improved the article. Please find the detailed responses below and the corresponding revisions/corrections highlighted/in track changes in the re-submitted files.]

Comment 2: [First, I would like to see a clear message from this paper about what readers can apply to future clinical practice. The indications for PN in clinical practice do not always follow guidelines, but are influenced by the factors such as the continuation of chemotherapy and the wishes of the patient and their family.  I would like to see the authors provide more support and decision-making regarding whether the findings of this study are still recommended and how they differ from the guidelines.]

Response 2: [The clinical insights from our study find that the palliative care pathways vary markedly between patients, making the judgement of the appropriateness of PN difficult. We however agree with you that it is appropriate to address this specifically in the paper. We have added a section in the discussion, under the paragraph 4.2 Clinical Implications, p. 11. line 365-375.]

Comment 3: [The study has included cases from 2011 to 2017, so the data do not reflect subsequent recent advances in cancer chemotherapy and treatment systems. I would also like to see the authors consider the possibility that the situation may be even different today, given the emergence of ICIs.]

Response 3: [Agree. We have, accordingly, included a paragraph with a reflection on the relevance of the data with recent advancements in anti-cancer treatment to emphasize this point. Changes can be found under the discussion section, in the strengths and limitations paragraph (4.5). page 13, line 428-438.]

Comment 4: [Clinical dietitians were reported to be involved in the care in 46% of cases (P6, L1-2). I would like to see a comparison of PN implementation situation between those with and without such involvement by dieticians (sub-analysis).  This could provide valuable data that will shed light on the role of registered dietitians in nutritional management.]

Response 4: [We agree that more emphasis could be placed on the involvement of the clinical dietitian. When doing sub-group analysis, we did not find particularly different treatment patterns of patients, other than that they were the once receiving higher doses for shorter periods of time, and received treatment for longer time periods. We chose to emphasize the involvement of the dietitian for those with the indication to resume anticancer treatment (who were able to resume treatment), and included more depth to the role of the dietitian in the discussion, as warranted in your suggestion for improvement. The included information can be found in the discussion section, now under paragraph 4.2. Clinical Implications, p. 10-11.]

Additional clarifications: [In addition, we have made adjustments according to the comments from reviewer 2. We have clarified and enhanced the readability of the tables and made some spelling and grammar improvements throughout the article. All changes can be found in the resubmitted file with track changes.]

Reviewer 2 Report

Comments and Suggestions for Authors

First, I would like to thank you for the opportunity to review the article “Parenteral Nutrition in Patients with Incurable Cancer: Exploring the Heterogeneous and Non-randomised Clinical Landscape.” Congratulations, and I would like to offer some suggestions for improving the article. In the abstract, I would summarize some of the numerical results presented and suggest including the clinical implications of your work in the conclusion. Regarding the introduction, I would summarize the information concerning the lack of evidence and heterogeneity and advise concluding the introduction with a clear objective statement about the purpose of the study. I suggest a sentence similar to: “The aim of the study is to describe the practices and actual outcomes of parenteral nutrition in patients with incurable cancer in Norwegian hospitals.”

In the abstract, I would summarize the information regarding the lack of evidence and heterogeneity and advise concluding the introduction with a clear objective statement about the purpose of the study. I believe the methodology section should include a more detailed description of the inclusion criteria, as well as indicate the limitations of the retrospective study. Similarly, section 2.2 should specify whether quality control measures were implemented for data collection (review by two independent reviewers, etc.). The results are clear, including adequate information on dosage, etc., and with a large number of participants. I would recommend performing a comparative statistical analysis between cancer types, etc., as well as improving the interpretation of the figures, such as Figure 2.A, which in lines 208 and 209 only states "illustrates the relationship between patients' body weight and median energy (kcal) provided through PN."
The discussion is comprehensive and includes references to future implications and ethical considerations. I believe it could be improved by including subheadings such as "Clinical implications," "Comparison with previous studies," and "Future research." Regarding the limitations, it is important to acknowledge the design limitations and incomplete documentation. I believe it could be improved by including a possible comparison with hospitals in other countries, given its focus on Norwegian hospitals.
The conclusions are consistent with the objectives, although I recommend including a clear clinical recommendation, such as the need to evaluate the indications for neuromuscular blocking (NMB), assessing its effectiveness in the care of people with incurable cancer.

Author Response

Comment 1: [First, I would like to thank you for the opportunity to review the article “Parenteral Nutrition in Patients with Incurable Cancer: Exploring the Heterogeneous and Non-randomised Clinical Landscape.” Congratulations, and I would like to offer some suggestions for improving the article.]

Response 1: [Thank you very much for taking the time to review this manuscript. Please find the detailed responses below and the corresponding corrections in track changes in the re-submitted files.]

Comment 2: [In the abstract, I would summarize some of the numerical results presented and suggest including the clinical implications of your work in the conclusion. Regarding the introduction, I would summarize the information concerning the lack of evidence and heterogeneity and advise concluding the introduction with a clear objective statement about the purpose of the study. I suggest a sentence similar to: “The aim of the study is to describe the practices and actual outcomes of parenteral nutrition in patients with incurable cancer in Norwegian hospitals.”]

Response 2: [Thank you. We have changed the abstract in accordance with your suggestions. The changes can be found in the Abstract on page 2.]

Comment 3: [In the abstract, I would summarize the information regarding the lack of evidence and heterogeneity and advise concluding the introduction with a clear objective statement about the purpose of the study.]

Response 3: [Thank you for this feedback. We have rewritten the aim at the end of the introduction, clearly stating the purpose of the study. Changes can be found in the end of 1. Introduction, page 3.]

Comment 4: [I believe the methodology section should include a more detailed description of the inclusion criteria, as well as indicate the limitations of the retrospective study. Similarly, section 2.2 should specify whether quality control measures were implemented for data collection (review by two independent reviewers, etc.).] 

Response 4: [Agree. We have, accordingly, modified the paragraph to emphasize these points. We have addressed the retrospective design and quality control measures. The research data manual and monitoring plan can be shared and published as a supplementary on request. Changes can be found in the method section, paragraph 2.2, page 4.]

Comment 5: [The results are clear, including adequate information on dosage, etc., and with a large number of participants. I would recommend performing a comparative statistical analysis between cancer types, etc., as well as improving the interpretation of the figures, such as Figure 2.A, which in lines 208 and 209 only states "illustrates the relationship between patients' body weight and median energy (kcal) provided through PN."]

Response 5: [Thank you for the thorough feedback. We agree regarding figure 2 and have, accordingly, improved the interpretation of the figure by explaining some key results in the text, as well as modifying the descriptive text in the figure and figure text to enhance readability. Changes can be found in the result section, paragraph 3.2.2., p 7-8.

Regarding the comparative statistical analysis, we have done several explorative comparisons, both on cancer type, treatment/no treatment, survival, length of parenteral nutrition treatment and so on, without finding very clear differences based on the characteristics of the patients. We therefore chose to describe the results from the overall population sample and not include the sub-group results in the article.]

Comment 6: [The discussion is comprehensive and includes references to future implications and ethical considerations. I believe it could be improved by including subheadings such as "Clinical implications," "Comparison with previous studies," and "Future research." Regarding the limitations, it is important to acknowledge the design limitations and incomplete documentation. I believe it could be improved by including a possible comparison with hospitals in other countries, given its focus on Norwegian hospitals.]

Response 6: [Thank you. We agree with the subheadings and have included these in accordance with your suggestion. Regarding the limitations, this is a good point. We have added a paragraph discussing this in the limitation section, and hope this better addresses the design and population limitations. Changes can be found in the discussion, under 4.5, strengths and limitations. p. 12.]

Comment 7: [The conclusions are consistent with the objectives, although I recommend including a clear clinical recommendation, such as the need to evaluate the indications for neuromuscular blocking (NMB), assessing its effectiveness in the care of people with incurable cancer.]

Response 7: [We agree with the suggestion to include a clear clinical recommendation. We chose to clarify the last part of the conclusion, emphasizing the need to study the patients who actually receive parenteral nutrition in clinical practice, to better be able to evaluate the indications and effectiveness in the care of patients with incurable cancer. Changes can be found in 5. Conclusion. p. 13.]

Additional clarifications: [In addition, we have made adjustments according to the comments from reviewer 1. We have clarified and enhanced the readability of the tables and made some spelling and grammar improvements throughout the article. All changes can be found in the resubmitted file with track changes.]

Round 2

Reviewer 2 Report

Comments and Suggestions for Authors

The authors have implemented or justified the recommendations given, and I believe it meets the journal's standards.